# The Epidemiology of Benign Proliferative Processes of the Skeletal System in Children

**DOI:** 10.3390/ijerph18179338

**Published:** 2021-09-03

**Authors:** Michal Rutkowski, Kinga Niewinska

**Affiliations:** 1Department of Trauma and Orthopaedic Surgery, Marciniak Lower Silesian Specialist Hospital, 54-049 Wroclaw, Poland; 2Department of Traumatology and Emergency Medicine of Developing Age, Wroclaw Medical University, 50-367 Wroclaw, Poland; kinga.niewinska@umed.wroc.pl

**Keywords:** osteochondroma, bone tumour, bone cyst, biopsy

## Abstract

A suspicion of a proliferative bone lesion in a child seems to be a major diagnostic problem for clinicians. There are no diagnostic and treatment algorithms described in the literature and no reliable cohort epidemiological data. Our study was conducted among 289 paediatric patients (0–18 years old) with an initial diagnosis of a bone tumour or tumour-like lesion. The study comprised a retrospective epidemiological analysis, an assessment of the concordance of the initial diagnoses with the histopathological diagnoses and an analysis of the specific locations of the various bone lesions. The results obtained have made it possible to formulate the following conclusions. (1) The most common proliferative bone lesion in children is osteochondroma; also common are fibrous dysplasia, non-ossifying fibromas and bone cysts. (2) Verifying the initial diagnosis by means of biopsy is essential. (3) Osteochondromas are typically located in the metaphyses of long bones, fibrous dysplasia in the femur and skull, cyst-like lesions in the proximal humerus and non-osteochondral fibromas exclusively in the lower limbs. What could improve the quality of treatment for children with primary proliferative bone diseases is the establishment of centres of paediatric orthopaedic oncology skilled in early diagnosis and prompt management.

## 1. Introduction

Proliferative bone lesions in children have always been a challenge for doctors, with regard to both diagnosis and treatment [1]. This medical problem, which lies at the interface between oncology, orthopaedics and paediatric surgery, requires multidisciplinary management in order to optimise treatment outcomes. 

The widespread occurrence of benign proliferative bone lesions in children significantly exceeds the number of diagnosed malignant lesions [2]. These lesions are frequently asymptomatic and diagnosis is often made accidentally. There are no available epidemiological data on the incidence of benign bone tumours in children. This is probably due to the often asymptomatic nature of the diseases and the fact that their incidences do not have to be recorded. Common benign bone lesions in children include solitary bone cysts, aneurysmal bone cysts, enchondromas, osteochondromas, non-ossifying fibromas and osteoid osteomas [3].

The objectives of the study were to:Carry out an epidemiological analysis of paediatric patients with diagnosed proliferative bone diseases;Assess the concordance between initial diagnoses and histopathological diagnoses;Analyse the locations of the tumours in the skeletal system.

## 2. Materials and Methods

The study was conducted among patients aged between 0 and 18 at the Department of Paediatric Surgery, Marciniak Lower Silesian Specialist Hospital, Emergency Medicine Centre in Wrocław. The data were obtained from the hospital’s AMMS computer system, including patients’ records as well as the department’s archives of histopathology results. The analysis covered patients hospitalised between 1 January 2015 and 31 December 2017 with initial diagnoses of bone tumours and tumour-like lesions. The patients examined in the study had the following initial diagnoses as defined in the International Statistical Classification of Diseases and Related Health Problems (ICD–10): D16—benign neoplasms of bone and articular cartilage; M85—other disorders of bone density and structure.

The following data were analysed:Initial and final diagnoses of the hospitalised patients;Concordance between initial and biopsy diagnoses;Patient age and sex distribution in the analysed population;Locations of neoplastic lesions in the skeleton;Detailed locations of tumours within the main groups of histopathological diagnoses.

### Statistical Analysis

The results obtained were subjected to a statistical analysis. The following parameters were calculated for all groups: number of cases (N), median (M), range (min–max) and lower and upper quartiles (25Q–75Q) of the analysed continuous parameters. Owing to a lack of variance homogeneity in the groups (variance homogeneity was checked by means of Bartlett’s test), verification of the hypothesis concerning the equality of the means of the various samples was performed with the non-parametric Mann–Whitney U test (for two groups) or the Kruskal–Wallis rank sum test (for three or more groups). For discrete parameters, the frequency of a trait across groups was analysed by means of the c2df test with Yates’ correction with an appropriate number of degrees of freedom df (df = (m − 1) * (n − 1), where m is the number of rows and n is the number of columns). *p* < 0.05 was regarded as statistically significant, while 0.05 < *p* < 0.1 was regarded as a possibility of a trend. The statistical analysis was conducted by means of the Microsoft Office Excel^®^ and EPIINFO Ver. 7.1.1.14 statistical software.

## 3. Results

### 3.1. Age and Sex Structure

The analysed population comprised 289 patients treated at the Department of Paediatric Surgery, T. Marciniak Lower Silesian Specialist Hospital, Emergency Medicine Centre in Wrocław, hospitalised between 1 January 2015 and 31 December 2017. Boys (178) constituted 61.6% and girls (111) 38.4% of the population. 

The average age of the patients was 11.4 years (SD = 4.1). The youngest patient was less than 1 and the oldest almost 18 years old.

### 3.2. Initial and Final Diagnoses

The patients examined in the study had the following initial diagnoses according to the International Statistical Classification of Diseases and Related Health Problems (ICD–10):D16—benign neoplasms of bone and articular cartilage;M85—other disorders of bone density and structure.

The available data on the first manifestations of bone tumours indicate that pain and swelling are the basic and most commonly observed symptoms. Pain usually involved a specific area; it was not strictly limited and was often associated with a minor injury to the area in question. Swelling was localised within joints in the vicinity of the bones affected by the disease. Other symptoms found in patients in whom a proliferative disease was eventually diagnosed included coldness of limb, thickening, a nodule detected by the parents or present from birth, palpable deformation of the cortical bone, deformation of the nail plate, pain and swelling of the toe (initially diagnosed as a reaction to a foreign body).

There was a large group of patients in whom the initial diagnosis of bone tumour was prompted by an X-ray taken due to an injury to another area.

The most common initial diagnoses are presented below (Figure 1).

The descriptive initial diagnoses based on analyses of imaging tests and the clinical presentation of the patients were as follows:Osteochondroma (n = 97);Cyst (n = 52);Tumour (n = 49);Fibrous dysplasia (n = 29);Non-ossifying fibroma (n = 28);Osteoid osteoma (n = 7);Synovial sarcoma (n = 5);Chondroma (n = 5);Benign chondroblastoma (n = 4);Osteolytic lesion (n = 4);Bone island (n = 2);Chondrosarcoma (n = 2);Osteosarcoma (n = 2);Ewing sarcoma (n = 2).

Surgical biopsy was performed in 234 patients. In 22% of cases (n = 63), the result of the biopsy corresponded to bone tissue without pathological lesions. In the remaining cases, the following final diagnoses were made (Table 1, Figure 2):

### 3.3. Concordance between Initial and Final Diagnoses

The concordance between the initial and final diagnoses in the various groups was analysed. The analysis encompassed patients with initial diagnoses defined by specific diseases. Patients with initial diagnoses such as tumour, tumour-like lesion, etc., were excluded. The result is illustrated by Table 2.

The various biopsy diagnoses were characterised by different levels of concordance with the initial diagnosis (c24 = 57.1, *p* = 0.00000). Cysts and osteochondromas were marked by the highest concordance. The biggest number of diagnostic errors occurred in non-ossifying fibromas suspected in the clinical material, and in other, less common proliferative bone processes.

### 3.4. Age and Sex Structure in Specific Histopathological Diagnoses

The sex distribution in specific histopathological diagnoses is presented in the figure below (Figure 3):

There were statistically significant differences in sex distribution between the various biopsy results (c24 = 14.64, *p* = 0.00552). Bone cysts, osteochondromas and non-ossifying fibromas occurred more frequently in boys, while fibrous dysplasia was diagnosed more often in girls. Other bone pathologies in the analysed material occurred with similar frequency in both sexes.

Patients’ ages at the time of biopsy in the specific histopathological diagnosis groups were statistically significantly different (*p* = 0.0270). This is illustrated by the following chart (Figure 4).

The average patient age at the time of diagnosing fibrous dysplasia and bone cysts was around 9 years. Patients with non-ossifying fibromas, osteochondromas and other tumour-like bone lesions were older and at the time of diagnosis were 12–13 years old.

### 3.5. Locations of Bone Tumours

In the analysed group of patients, tumours or tumour-like lesions were located in various bones of the skeleton. The locations of lesions on the basis of the available data are presented in Table 3.

#### 3.5.1. Locations of Osteochondromas

The biggest group of histopathological diagnoses in the analysed group of patients comprised osteochondromas. The most frequent locations of osteochondromas were long bones (shin bones—38.3% (n = 31), femur—22.3% (18), humerus—13.6% (n = 11)) (Table 4). Multiple lesions were diagnosed in 6/81 patients (7%).

#### 3.5.2. Locations of Fibrous Dysplasia

In nearly half of the cases (n = 5), fibrous dysplasia was located in the bones of the skull (mandible, parietal bone and frontal bone). The second most common location was the femur (n = 3). Two lesions were found in the distal metaphysis and one in the proximal epiphysis of the femur. There were isolated cases of lesions in other locations in the analysed material: pelvis (n = 1), collar bone (n = 1) and tibial diaphysis (n = 1). Among the patients with fibrous dysplasia in long bones, the dominant group were girls (3:1); likewise, girls were the dominant group in the case of lesions located in cranial bones (3:2).

#### 3.5.3. Locations of Bone Cysts

Of the lesions found in the analysed histopathological material, 54.5% (n = 6) were aneurysmal cysts, and 45.5% (n = 5) were solitary bone cysts. Owing to the difficulty of differentiating between these lesions on the basis of imaging tests and clinical examination, a decision was made to include both diagnoses in one group in further analysis. All cysts in the analysed material occurred in long bones. They were most prevalent in the humerus (n = 6). The other affected bones in the studied population were the proximal femur (n = 2), distal tibial metaphysis (n = 1), tibial diaphysis (n = 1) and distal radial metaphysis (n = 1). 

#### 3.5.4. Locations of Non-Ossifying Fibromas

In the analysed material, all the cases of diagnosed non-ossifying fibromas confirmed by histopathological tests were located in the long bones of the lower limbs: 64% (n = 9) in the lower leg and 36% (n = 5) in the femur. 

#### 3.5.5. Locations of Other Tumours

The locations of other tumours and tumour-like bone lesions in patients undergoing histopathological examination varied. Most lesions (59%) were located in long bones (72% of them affected lower limb bones and 28% upper limbs). A total of 19% of the lesions were located in the small bones of the hand and foot, and the remaining 22% affected other bones, including the ribs, scapula, clavicle, skull, pelvis and spine.

### 3.6. Summary of the Results

Proliferative bone lesions in children are diagnosed more frequently in boys (61.6%) than in girls (38.4%). This result is statistically significant (chi2 = 6.755, *p* = 0.009).

In the analysed material, bone cysts confirmed by histopathology occurred only in boys. The average age of patients at the time of diagnosis was 11 years. Only in patients with fibrous dysplasia did females (63.64%) predominate over males (36.36%); however, this is without statistical significance (chi2 = 2.068, *p* = 0.15).

An analysis of histopathological diagnoses in the studied population has shown that the most common proliferative lesions in the skeletal systems of children are osteochondromas (47%), non-ossifying fibromas (8%), fibrous dysplasia (7%) and bone cysts (6%).

Concordance between the initial and histopathological diagnoses was high (98.77%) in osteochondromas. In the case of these lesions, a one-stage procedure (excisional biopsy) appears to be the right therapeutic option.

Osteochondromas are located mainly in long bones (shin bones, femur and humerus).

Diagnosed non-ossifying fibromas occurred exclusively in the long bones of the lower limbs: 64% (n = 9) in the shin bones and 36% (n = 5) in the femur.

In 45.5% of cases, fibrous dysplasia was located in the cranial bones, with the femur being the second most frequent location. 

Bone cysts occur most often in long bones, with the humerus being the most common location. Significantly, there is a high level of disparity between the initial and final diagnoses in this group of patients.

In the case of rarer kinds of tumours, biopsy results are usually different from the suspected clinical diagnosis. This is why biopsy-based verification prior to the final treatment is still the method of choice in the treatment of bone tumours.

## 4. Discussion

Despite the fact that extensive effort towards the proper diagnosing, classifying and treating of bone tumours has been going on in medicine since the 1920s, the problem still remains a difficult challenge for clinicians, especially when the patient is a child [4]. There are no structured treatment algorithms in the literature, and each case should be treated individually. 

The main problem in diagnosing both benign and malignant bone tumours is their variable progression. In their early stages, the diseases are often asymptomatic. Patients with early symptoms frequently do report to their doctors, but a lack of characteristic symptoms delays the right initial and further diagnoses [5].

It is not uncommon for a tumour to be diagnosed in the course of a pathological fracture as a result of a radiological diagnosis of an injury to the affected area. This is why it is so important to carry out additional tests in patients in whom a typical treatment fails to bring the desired therapeutic outcome; however, this was not an aim of our study.

Regarding the symptoms displayed by the patients in the analysed group, there was a striking multiplicity of symptoms (pain, palpable mass, swelling), but despite this, a large proportion of bone tissue pathology was diagnosed during imaging tests performed for other reasons, usually due to an injury. 

An additional difficulty in diagnosing these conditions is their rarity. The analysed group of patients (n = 289) made up less than 1% of patients hospitalised at the Department of Paediatric Surgery, Marciniak Lower Silesian Specialist Hospital in Wrocław during the studied period. Apparently, the oncological vigilance of a clinician examining the patients can make it possible to diagnose or exclude a bone tumour.

Due to the rarity of diagnoses, patients with bone tumours should be referred to centres specialising in paediatric orthopaedic oncology. 

In the analysed clinical material, proliferative bone diseases were more frequent among boys (61.6%), with the average age of patients at the time of diagnosis and commencement of treatment being 11 years. The dominance of males among patients was also confirmed by a study based on an analysis of the Indian population. In this study, the average age of patients was 13 years [1]. It should be noted that in the analysed material, bone cysts confirmed by histopathology occurred only in boys. Data from the literature point to a strong male dominance for simple cysts (3:1) and no significant sex differences in the incidence of aneurysmal cysts [2,6].

The strategy for treating osteochondromas remains controversial, with some authors recommending observation of asymptomatic lesions, and others advocating early surgical intervention to prevent skeletal deformation [7].

Routine removal of all diagnosed osteochondromas does not seem to be the right course of action. It is essential to remove symptomatic lesions causing skeletal deformities as well as those that increase in size during the follow-up period [8]. An important issue is the optimisation of the time when surgical intervention should be undertaken. It seems that in the case of isolated asymptomatic osteochondromas located in the area of the growth, plate surgical treatment can be postponed until bone growth is finished. An earlier surgical intervention in such a location may lead to impaired limb growth as a result of iatrogenic damage to the growth plate, or, in the case of incomplete resection—due to a fear of damage—to recurrence. Postponing surgical treatment could lead to fewer recurrences and thus fewer secondary surgical interventions.

According to the data found in the literature, malignant transformations of isolated osteochondromas into chondrosarcomas are rare (<1%), always produce symptoms such as pain and/or an increase in tumour mass and occur after adolescence. In the case of multiple lesions, the risk of malignant progression before the end of bone growth has been estimated at approximately 1–4%. Therefore, in patients with multiple lesions, early surgical intervention should be the treatment of choice [2,9,10].

Given the high degree of concordance between the initial diagnosis based on radiological findings and the biopsy diagnosis (99%), a one-stage biopsy combined with a complete resection of symptomatic osteochondromas (excisional biopsy) seems to be an appropriate procedure.

Another important group of diagnoses in the analysed material were patients with bone diseases which often manifest themselves in the form of a tumour or thickening of a specific area, but in which the underlying process is not neoplastic. Tumour-like lesions in the musculoskeletal system are an important issue in paediatric orthopaedics due to their high incidence. Although these lesions are not life-threatening, the detection of a “tumour” is often a major psychological challenge, especially for the parents of a paediatric patient.

Owing to the complex and not yet fully known pathogenesis of these conditions, there is great controversy over the treatment of such patients. In the analysed material, 109 patients were hospitalised due to suspected tumour-like lesions of the bone (bone cyst—n = 52, fibrous dysplasia—n = 29, non-ossifying fibroma—n = 28). Given the typical progression of the disease, most patients underwent conservative treatment. 

Another fairly large group of tumour-like lesions in the analysed material were non-ossifying fibromas. They were located only in the metaphyses of the lower limb long bones, mainly in the lower leg bones. According to the literature, growth plates in the distal part of the femur as well as distal and proximal metaphyses of the tibia are typical locations of non-ossifying fibromas [11]. The disease is usually asymptomatic and self-limiting, and in most cases the lesions resolve spontaneously. The condition is usually diagnosed accidentally, when X-rays are taken because of limb injuries. If the dimension of an asymptomatic lesion does not exceed 50% of the bone diameter, periodic X-ray checks are recommended [12]. 

Pathological fractures within a non-ossifying fibroma are rare (there was one pathological fracture within a non-ossifying fibroma in the analysed group of patients). Twenty-three cases of pathological fractures within histologically verified non-ossifying fibromas were described in the Mayo Clinic experience over a period of nearly 50 years. In all these cases, the lesions occupied more than 50% of bone diameter in both the anteroposterior and the lateral radiographs. All fractures (except one) were located in the lower limbs, most often in the distal end of the tibia. Most authors advocate conservative treatment of pathological fractures within benign bone tumours until osteosynthesis is achieved, followed by treatment of the underlying disease depending on its clinical presentation, with fractures within non-ossifying fibromas having the best prognosis for achieving synostosis [13,14]. 

For lesions exceeding 50% of the bone diameter, surgical treatment in the form of curettage and bone grafting should be considered due to an increased risk of pathological fracture [8]. In addition, larger lesions may lead to secondary aneurysmal bone cysts [12].

Biopsy is an essential element in the diagnosis of most bone tumours. It should be performed after a full initial diagnosis, including appropriate diagnostic imaging. Biopsy makes it possible to establish a histopathological diagnosis and to assess prognostic factors [15]. The preferred method is surgical biopsy carried out at a centre experienced in comprehensive treatment of bone tumours. In Mankin’s study, authors compared patients in whom a biopsy was performed in a non-specialised institution with patients in whose case the biopsy was carried out in centres treating bone tumours. Errors, complications and changes in the course and outcome of the treatment were two to twelve times greater (*p* < 0.001) when the biopsy was performed in a referring institution instead of in a treatment centre [16].

What is striking in the analysed material is the large group of negative biopsies. A histology result showing trabeculae without atypical features occurred in 22% (n = 63) of cases. This may have stemmed from lesions diagnosed in imaging tests but not being, in fact, proliferative in nature, or from errors during sample taking in the course of surgical biopsies. When material is sampled for histopathological examination, what needs to be taken into account is the fact that the central part of the tumour mass usually contains necrotic tissue without tumour cells. If only this tissue is sampled, the histopathological diagnosis is chronic osteomyelitis. Bone tumours contain a peripheral reaction zone with bone tissue breakdown and formation, but without malignancy traits. Taking samples from this zone may also lead to a misdiagnosis. The size of the sample must make it possible to carry out a correct histological assessment (at least 1.0 cm × 1.0 cm × 1.0 cm) [17]. 

In the case of bone cysts, especially when planning injection treatment, biopsy is often omitted from the diagnostic process. In the case of aneurysmal cysts, biopsy is essential. It can be carried out by means of a trocar or, even better, a surgical procedure, or in combination with curettage (“curopsy”). The procedure is performed to exclude telangiectatic osteosarcoma and giant-cell tumour of bone, as well as to differentiate between an aneurysmal cyst and a simple cyst. A simple cyst can be suspected if clear fluid is aspirated and the cavity is completely opaque in a cystographic examination. However, clear fluid may also be obtained from aneurysmal cysts, though the cavity is without complete opacity in this case. Blood-stained fluid may still indicate a simple cyst, especially in the case of an injury, and even the co-existence of a simple and an aneurysmal cyst. In the case of some lesions, especially in the spine, aneurysmal cyst biopsy may be associated with intraoperative bleeding, which can be avoided by means of prior embolisation [6].

Surgical biopsy appears to be the correct treatment procedure in the case of suspected bone cysts, especially given the fact that by compromising cyst wall integrity it is possible to induce the remodelling of the cyst cavity and combine it with curettage. The analysis results reveal a significant percentage of diagnostic errors in lesions initially diagnosed as bone cysts.

The symptoms of both benign and malignant bone tumours are not specific. There are many reports in the literature of delayed diagnosis of cancer as a result of incomplete diagnostic imaging, and of patients being operated on as a result of a misdiagnosis. Diagnostic errors also occur in patients on whom a range of additional tests were performed. This stems from the insidious nature of proliferative diseases rather than from diagnostic omission [18,19].

The task of a physician is to assess, on the basis of an anamnesis, whether the patient’s complaints are commensurate with the potential damage resulting from an injury. In the case of bone tumours, an injury is a factor drawing attention to the ongoing pathology and not the actual cause of the complaint.

## 5. Conclusions

The most common proliferative bone lesion in children is osteochondroma. Other common bone diseases in this age group include fibrous dysplasia, non-ossifying fibromas and bone cysts.

In the case of bone cysts, the concordance between histological diagnosis and initial diagnosis was complete. However, the initial diagnosis of bone cysts was overused in the diagnosis of tumour-like bone lesions and was confirmed by biopsy in only 1/3 of cases. Concordance between the biopsy result and the initial diagnosis of fibrous dysplasia occurred in 2/3 of cases, and in the case of non-ossifying fibromas in 3/5 of cases. Verification of the initial diagnosis by means of biopsy is relevant in the case of proliferative bone lesions in children.

Proliferative lesions are most commonly located in the metaphyses and diaphyses of long bones, depending slightly on the subtype of the lesion.

## Figures and Tables

**Figure 1 ijerph-18-09338-f001:**
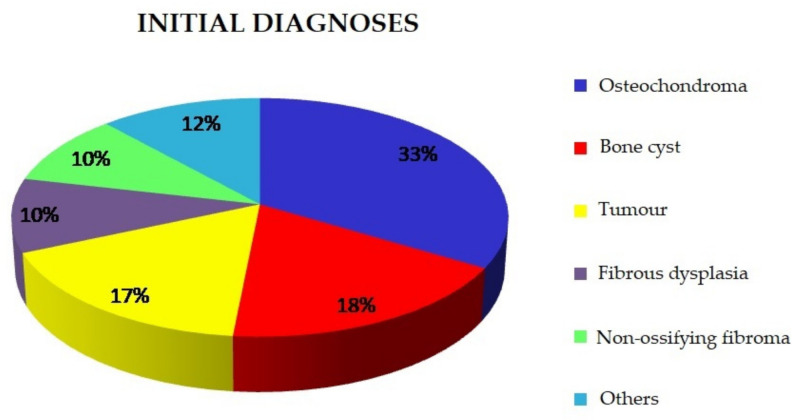
Initial diagnoses.

**Figure 2 ijerph-18-09338-f002:**
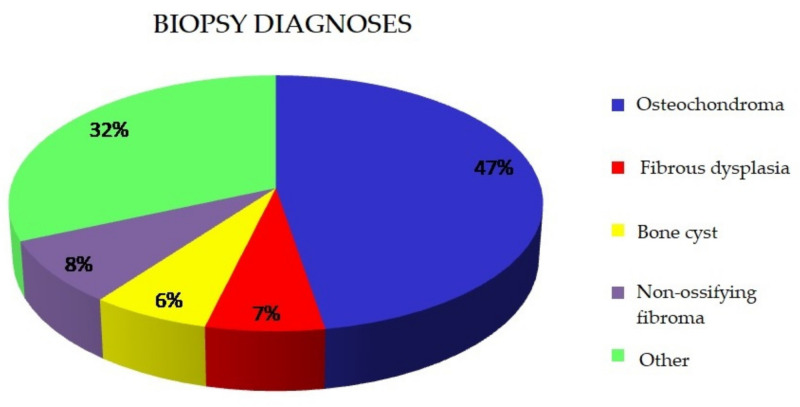
Biopsy diagnoses.

**Figure 3 ijerph-18-09338-f003:**
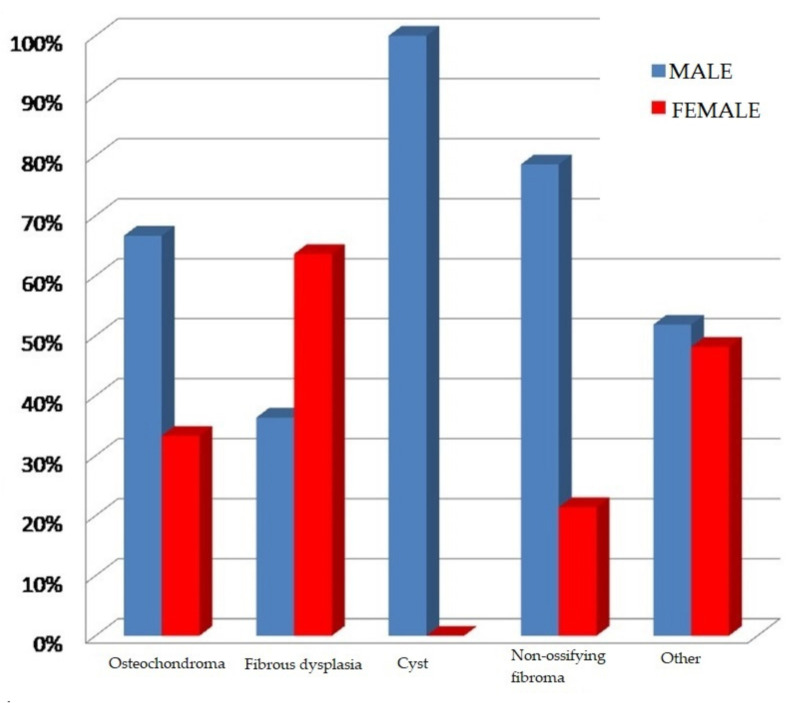
Sex distribution in the various histopathological diagnoses.

**Figure 4 ijerph-18-09338-f004:**
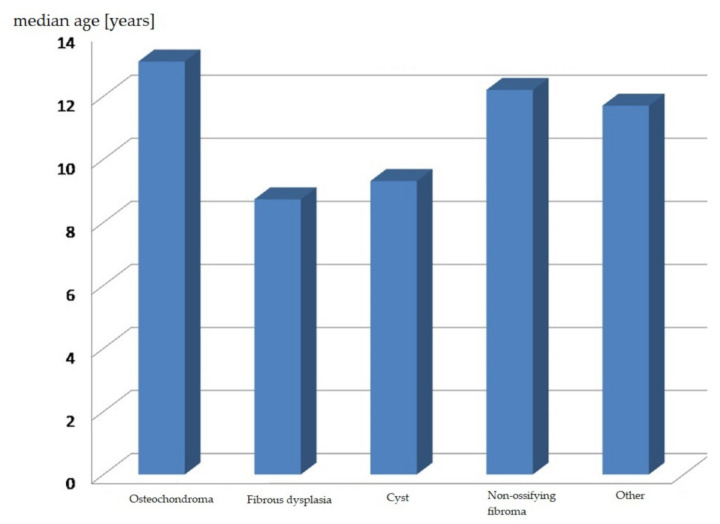
Age distribution in specific histopathological diagnoses.

**Table 1 ijerph-18-09338-t001:** Biopsy diagnoses.

Biopsy Diagnosis	Number	Percentage
Osteochondroma	81	47.4
Fibrous dysplasia	11	6.4
Bone cyst	11	6.4
Non-ossifying fibroma	14	8.2
Other	54	31.6

**Table 2 ijerph-18-09338-t002:** Concordance between initial and final diagnoses.

Biopsy Diagnosis	No Concordance between Initial and Biopsy Diagnoses	Concordance between Initial and Final Diagnoses	Total
Osteochondroma	1	80	81
	1.23%	98.77%	
Fibrous dysplasia	2	5	7
	28.57%	71.43%	
Cyst	0	11	11
	0.00%	100.00%	
Non-ossifying fibroma	5	8	13
	38.46%	61.54%	
Other	18	12	30
	60.00%	40.00%	
Total	26	116	142

**Table 3 ijerph-18-09338-t003:** Locations of lesions in the skeleton.

Biopsy Diagnoses	Cranial Bones	Pelvis Spine	Scapula Clavicula RibRibs	Hand	Foot	Lower Leg	Forearm	Humerus	Femur
Osteochondroma	0	3	5	1	9	31	3	11	18
	0.00%	3.70%	6.17%	1.23%	11.11%	38.27%	3.70%	13.58%	22.22%
Fibrous dysplasia	5	1	1	0	0	1	0	0	3
	45.45%	9.09%	9.09%	0.00%	0.00%	9.09%	0.00%	0.00%	27.27%
Cyst	0	0	0	0	0	2	1	6	2
	0.00%	0.00%	0.00%	0.00%	0.00%	18.18%	9.09%	54.55%	18.18%
Non-ossifying fibroma	0	0	0	0	0	9	0	0	5
	0.00%	0.00%	0.00%	0.00%	0.00%	64.29%	0.00%	0.00%	35.71%
Other	5	2	5	6	4	13	2	7	10
	9.26%	3.70%	9.26%	11.11%	7.41%	24.07%	3.70%	12.96%	18.52%
Total	10	6	11	7	13	56	6	24	38

**Table 4 ijerph-18-09338-t004:** Locations of osteochondromas.

Total	Diaphysis	Metaphysis	Epiphysis	Location
1	0	0	1	Foot
31	0	31	0	Lower leg
3	0	2	1	Forearm
11	2	9	0	Arm
18	0	17	1	Upper leg
64	2	59	3	Total

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
