# Peer review of "The Epidemiology of Benign Proliferative Processes of the Skeletal System in Children"

_ijerph, 2021, doi:10.3390/ijerph18179338_

Round 1

Reviewer 1 Report

This study analysed 289 paediatric patients with an initial diagnosis of a bone tumour or tumour-like lesion, patients’ characteristics were described and concordance between initial diagnoses and histopathological diagnoses were assessed.

The sample of this study is very specific, i.e. they were patients of initial diagnosis of a bone tumour or tumour-like lesion. The authors may provide some information on the diagnosis of Osteochondroma, Fibrous dysplasia, Bone cyst among the general young population.

The authors may conduct sensitivity and specificity analysis when assessing the concordance between the initial diagnoses and histopathological diagnoses.

Author Response

Dear reviewer,

Thank you very much for the review within such a short time.

Sensitivity and specificity analysis cannot be performed, because number of false negative results is unknown. However a positive predictive value of the initial diagnoses was calculated and was PPV=0,82.

We hope our study was found interesting and valuable by you.

We revised the manuscript accordingly and its final version is also submitted.

Please do not hesitate to contact us if you have any further queries.

Yours sincerely

Dr Kinga Niewinska

Dr Michał Rutkowski

Reviewer 2 Report

Thank you for this interesting manuscript.

The manuscript contributes valuable knowledge.

In the abstract, I lack a definition of children. Is it 0-18 years?

Introduction: "Proliferative bone lesions in children have always been a challenge to doctors, both with regard to diagnosis and treatment." This sentence needs a reference. I would also like to know if there are similar challenges in all countries, or if the possibility of getting a diagnosis and treatment differ globally.

"The widespread occurrence of benign proliferative bone lesions in children significantly exceeds the number of diagnosed malignant lesions." This sentence needs a reference. I would also like to know if this is similar in all countries, or if it differs globally.

In the result, I think this text should be deleted “This section may be divided by subheadings. It should provide a concise and precise description of the experimental results, their interpretation, as well as the experimental conclusions that can be drawn.”

Table 4 needs to be corrected in the layout.

"Proliferative bone lesions in children are diagnosed more frequently in boys (61.6%) than in girls (38.4%).” Is there a statistical difference?

"Only in patients with fibrous dysplasia did females (63.64%) predominate over males (36.36%).” Is there a statistical difference?

Discussion: "Despite the fact that extensive effort for proper diagnosing, classifying and treating bone tumours has been going on in medicine since the 1920s " This sentence needs a reference.

Conclusion: I have no explanation for what knowledge this article contributes to. I would like a comparison between the results of this study and previous research.

Author Response

Dear reviewer,

Thank you very much for the review within such a short time. We are glad our study was found interesting and valuable by you. All comments and queries were invaluable for correction and improvement of our manuscript.

Please find our responses to your remarks. point-by-point below:

  • In the abstract, I lack a definition of children. Is it 0-18 years?

According to Polish law practice children are defined as individuals below the age of 18 years. Patients qualifying into this category are primarily treated in Paediatric Centers and very rarely transferred under the care of staff in adult Units. We have added this particular piece of information in the adstract.

  • Introduction: "Proliferative bone lesions in children have always been a challenge to doctors, both with regard to diagnosis and treatment." This sentence needs a reference. I would also like to know if there are similar challenges in all countries, or if the possibility of getting a diagnosis and treatment differ globally.

We have added a reference to a study from India. Authors describe 50 cases of primary tumors of bone in children.  They agree that diagnosis and treatment of these patients is difficult.

We performed a literature search in preparation for this study and there are not many obtainable data to compare. However the main difficulty is due to a relative lack of specific symptoms and physicians hesitance to overuse of diagnostic imaging during a developing age.

  • "The widespread occurrence of benign proliferative bone lesions in children significantly exceeds the number of diagnosed malignant lesions." This sentence needs a reference. I would also like to know if this is similar in all countries, or if it differs globally.

We have added a missing reference. WHO Classification of Tumours provides a general epidemiology of tumours.

  • In the result, I think this text should be deleted "This section may be divided by subheadings. It should provide a concise and precise description of the experimental results, their interpretation, as well as the experimental conclusions that can be drawn."

We have deleted this part added by mistake.

  • Table 4 needs to be corrected in the layout.

The corrected version is attached.

  • "Proliferative bone lesions in children are diagnosed more frequently in boys (61.6%) than in girls (38.4%)." Is there a statistical difference?

This result is statistically significant (chi2=6,755, p=0,009). We have added this piece of information in the manuscript.

  • "Only in patients with fibrous dysplasia did females (63.64%) predominate over males (36.36%)." Is there a statistical difference?

There is no statistical significance (chi2=2,068, p=0,15). We have added this piece of information in the manuscript.

  • Discussion: "Despite the fact that extensive effort for proper diagnosing, classifying and treating bone tumours has been going on in medicine since the 1920s " This sentence needs a reference.

We have added a missing reference. In 1920 J.C. Bloodgood, E.A. Codman, A. Kolodny and J. Ewing made efforts to establish the first database of bone tumors.

Clin Orthop Relat Res (2009) 467:2771–2782 DOI 10.1007/s11999-009-1049-6

  • Conclusion: I have no explanation for what knowledge this article contributes to. I would like a comparison between the results of this study and previous research.

The paper describes the frequency and occurrence of benign bone tumors in children and I believe our findings may be important for physicians. According to PubMed search we performed there have not been many studies, which would describe the specificity of skeletal tumours in different various age groups of patients below the age of 18 years. Musculoskeletal tumors in children and adolescent are relatively rare, thus our paper, which describes the condition epidemiology on a population of nearly 300 cases, provides a valuable tools for a diagnostic and management process.

We revised the manuscript accordingly and its final version is also submitted.

Please do not hesitate to contact us if you have any further queries.

Yours sincerely

Dr Kinga Niewinska

Dr Michał Rutkowski

Reviewer 3 Report

This is an interesting study conducted in a significant sample of 289 paediatric patients retrospectively. Patients had an initial diagnosis of a bone tumour or tumour-like lesion, with assessment of concordance of initial diagnoses with histopathological diagnoses, as well as analysis of the specific location of the various bone lesions detected.

Conclusions were also useful: most common proliferative bone lesion in children being osteochondroma; verifying initial diagnosis with biopsy was essential; osteochondromas typically located in metaphyses of long bones, fibrous dysplasia in femur and skull, cyst-like lesions in proximal humerus, and non-osteochondral fibromas exclusively in the lower limbs.

However, the conclusion: “One-stage resection appears to be an appropriate treatment procedure”, as stated in the abstract does not seem to be part of the Methods section and is neither supported in the Results section.

The number of subjects in Table 1 is 171 in total, whereas in Table 2 the number of subjects is 142 in total. Since in the text it is stated that surgical biopsy was obtained from 234 patients (-63 cases of bone lesions without pathological diagnosis)=171, Please check why table 2 only contains a fraction of those.

Figure 3: please avoid the terms Man/Woman, as these are not adult patients.

Discussion, lines 247-8: This may well be, however this article is not about treatment algorithms, as seen by the methodology and results sections; it is about diagnosis.

Author Response

Dear reviewer,

Thank you very much for the review within such a short time. We are glad our study was found interesting and valuable by you. All comments and queries were invaluable for correction and improvement of our manuscript.

Please find our responses to your remarks. point-by-point below:

  • "One-stage resection appears to be an appropriate treatment procedure", as stated in the abstract does not seem to be part of the Methods section and is neither supported in the Results section."

          We have deleted sentence.

  • The number of subjects in Table 1 is 171 in total, whereas in Table 2 the number of subjects is 142 in total. Since in the text it is stated that surgical biopsy was obtained from 234 patients (-63 cases of bone lesions without pathological diagnosis)=171, Please check why table 2 only contains a fraction of those.

In table 2 concordance between initial and final diagnoses in the various groups was analysed. The analysis encompassed patients with initial diagnoses defined by specific diseases. Patients with initial diagnoses like tumour, tumour-like lesion etc. were excluded. This information is included in the text.

Figure 3: please avoid the terms Man/Woman, as these are not adult patients.

We have changed terms for Male/female.

Discussion, lines 247-8: This may well be, however this article is not about treatment algorithms, as seen by the methodology and results sections; it is about diagnosis.

We have changed this sentence for: “That is why it is so important to carry out additional tests in patients in whom a typical treatment fails to bring a desired therapeutic outcome however this was not an aim of our study.”

We revised the manuscript accordingly and its final version is also submitted.

Please do not hesitate to contact us should you have any further queries.

Yours sincerely

Dr Kinga Niewinska

Dr Michał Rutkowski

Round 2

Reviewer 2 Report

The authors have revised the manuscript in accordance with the reviewer's comments.